# Otto Engine: Classical and Quantum Approach

**DOI:** 10.3390/e22070755

**Published:** 2020-07-09

**Authors:** Francisco J. Peña, Oscar Negrete, Natalia Cortés, Patricio Vargas

**Affiliations:** 1Departamento de Física, Universidad Técnica Federico Santa María, Casilla 110-V, Valparaíso 2390123, Chile; oscar.negrete@usm.cl (O.N.); natalia.cortesm@usm.cl (N.C.); patricio.vargas@usm.cl (P.V.); 2Centro para el Desarrollo de la Nanociencia y la Nanotecnología, Santiago 8320000, Chile

**Keywords:** thermodynamics, heat engines, quantum thermodynamics

## Abstract

In this paper, we analyze the total work extracted and the efficiency of the magnetic Otto cycle in its classic and quantum versions. As a general result, we found that the work and efficiency of the classical engine is always greater than or equal to its quantum counterpart, independent of the working substance. In the classical case, this is due to the fact that the working substance is always in thermodynamic equilibrium at each point of the cycle, maximizing the energy extracted in the adiabatic paths. We apply this analysis to the case of a two-level system, finding that the work and efficiency in both the Otto’s quantum and classical cycles are identical, regardless of the working substance, and we obtain similar results for a multilevel system where a linear relationship between the spectrum of energies of the working substance and the external magnetic field is fulfilled. Finally, we show an example of a three-level system in which we compare two zones in the entropy diagram as a function of temperature and magnetic field to find which is the most efficient region when performing a thermodynamic cycle. This work provides a practical way to look for temperature and magnetic field zones in the entropy diagram that can maximize the power extracted from an Otto magnetic engine.

## 1. Introduction

The classical standard nonmagnetic Otto cycle is widely used in present-day technology, as it is the thermodynamic cycle most commonly found in automobiles engines. This cycle consists of two classical isochoric and two classical adiabatic processes. In the isochoric processes, the system interacts with either of two thermal reservoirs at temperatures $T_{l}$ and $T_{h}$, with $T_{h}>T_{l}$, and each one of these processes is followed by a classical adiabatic process which enables work extraction. In this cycle, the results of the efficiency depend on the nature of the working substance (through its energy spectrum) and the contributions of work and heat are separate in its stages, which facilitates theoretical modelling. These characteristics favor its extension to its quantum version, which has been studied extensively in recent years [1,2,3,4,5,6,7,8,9,10,11,12,13,14,15,16,17,18,19].

Otto quantum cycle similarly consists of four processes: two quantum isochoric processes and two quantum adiabatic processes. The quantum isochoric process is very similar to the classical one in the sense that both admit changes in the temperature of the system trough heat exchange with zero work performed [20]. The quantum scenario requires that the energy-level populations of the working substance remain constant for each of the quantum states as the volume varies, ensuring that entropy remains unchanged. This implies that the quantum adiabatic processes should be realized in a time frame that avoids transitions between levels that can be generated while it is carried out. Deffner and Campbell define this difference [21]: “*Quantum adiabatic processes form only a subset of classical adiabatic processes*”. Therefore, the development of a classical and quantum Otto cycle differs not only in the time of their processes but also in the physical concepts involved. Moreover, it is essential to mention that the word “classical” should not be directly interpreted as if the working substance is “classical.” The notion of “classical” refers here to the four stages developed in the classical Otto cycle that are in thermodynamic equilibrium. On the other hand, the working substance of the quantum cycle reaches thermal equilibrium only in two stages of the cycle (the isomagnetic paths). Given the differences of the classical and quantum processes of the adiabatic Otto’s stage, two questions naturally arise: How are work performance and efficiency when these two approaches, classical and quantum, are applied to the same working substance? What conditions should fulfill the energy spectrum of the working substance in both approaches such that this difference is as small as possible?

In this work, we give a possible solution to these questions using as an example a magnetic Otto cycle under the classical and quantum thermodynamics formulations. In the magnetic Otto cycle, the two isochoric processes are replaced by two isomagnetic processes. Our first discussion entails a comparison between both thermodynamical approaches to explore why the classical work is greater than its quantum counterpart. We argue that the answer can be regarded as a consequence of a free energy principle for systems in thermal equilibrium. We analyze a particular case of a two-level system working under a magnetic Otto cycle, where we show that the total work extracted is equal under both formulations regardless of the magnetic dependence of the energy levels. Finally, we explore the consequences of including more energy levels and how this affects the behavior of thermal quantities and work as compared to the simple case of a two-level system.

## 2. Quantum and Classical Magnetic Otto Cycle: Classical and Quantum Work

To describe the Otto cycle classical and quantum formulation, we show a schematic representation of an entropy (*S*)–external field diagram in Figure 1. In our notation, *T* will refer to temperature and *B* to external magnetic field (both parameters in arbitrary units). The adiabatic and isomagnetic processes are represented in Figure 1 by horizontal and vertical lines, respectively.

The cycle is thermodynamically defined by the two heat reservoirs (hot and cold) and two extreme values of the external field, $B_{h}$ and $B_{l}$. Usually, the low temperature stage $\left(T_{l}\right)$ corresponds to a low external field value $\left(B_{l}\right)$ and the high temperature stage $\left(T_{h}\right)$ corresponds to the high external field value ($B_{h}$). This is not true for all working substances [22,23] because we need to observe the behaviour of the entropy as a function of different temperature and external field parameters in order to know which specific values correspond to each point in the cycle. As an example, the entropy behavior in quantum dots [22,23] is such that the points where the temperature reservoirs are located are crossed with the fields values. Nevertheless, the conclusions obtained in this work can be applied to this kind of system regardless of the reservoirs are located in the cycle.

The cycle presented in Figure 1 operates following the sequence A→B→C→D→A. The output power is defined as the work done per cycle divided by τ, with τ being the duration of each cycle iteration. It is important to point out that, under this formulation, thermal equilibrium is reached at the thermal reservoirs for both classical and quantum Otto cycles. For the first isomagnetic stroke B→C (see Figure 1) with a hot reservoir at temperature $T_{h}$ at point C, the working substance reaches the temperature of the hot reservoir. For the second isomagnetic process, from D to A (see Figure 1), the system is put into contact with a cold reservoir at temperature $T_{l}$ up to the working substance reaching the same temperature as the cold reservoir. Contrary are the cases of adiabats (classical and quantum case), where the systems are disconnected from the reservoirs and the external field is varied from $B_{l}$ to $B_{h}$ (process A→B) and vice-versa (process C→D). We use the superscript *q* for all quantum thermodynamics variables, while we use cs for the classical ones.

In the quantum scenario, the heat absorbed $\left(Q_{in}^{q}\right)$ and released $\left(Q_{out}^{q}\right)$ are given by the following [12,20,22,23]:(1)$$\begin{array}{c} Q_{in}^{q}=\sum_{s}E_{s}\left(B_{h}\right)\left[P_{s}^{C}\left(T_{h},B_{h}\right)-P_{s}^{B}\right], \end{array}$$
(2)$$\begin{array}{c} Q_{out}^{q}=\sum_{s}E_{s}\left(B_{l}\right)\left[P_{s}^{A}\left(T_{l},B_{l}\right)-P_{s}^{D}\right], \end{array}$$
where $E_{s}\left(B_{l(h)}\right)$ is the energy spectrum of the working substance evaluated in the low (high) temperature value in the cycle, $P_{s}$ corresponds to the probabilities of occupation along the cycle, and the index *s* represents the different quantum numbers that characterise a quantum state of the working substance. To satisfy the adiabatic nature of the process under the quantum formulation, one way is that the probabilities of occupation must satisfy the following conservation condition:(3)$$\begin{array}{c} P_{s}^{B}=P_{s}^{A}\left(T_{l},B_{l}\right),\;P_{s}^{D}=P_{s}^{C}\left(T_{h},B_{h}\right). \end{array}$$

Using these relations, we can define the total work per cycle:(4)$$\begin{array}{ccc} W^{q} & = & Q_{in}^{q}+Q_{out}^{q}\\ & = & \sum_{s}\left(E_{s}\left(B_{h}\right)-E_{s}\left(B_{l}\right)\right)\left[P_{s}^{C}\left(T_{h},B_{h}\right)-P_{s}^{A}\left(T_{l},B_{l}\right)\right]. \end{array}$$

Here, we can see one of the main difference between the classical and quantum approaches. While in classical thermodynamics we require the system to be in thermal equilibrium at every moment, here, we can see that, even if the points B and D do not fulfill this condition, the entropy conservation along the adiabatic paths can be made just with only two equilibrium points (A and C). Accordingly, the system reaches thermal equilibrium with the thermal reservoirs in the isochoric stages. This naturally means that the density matrix must be described as a thermal equilibrium state (Gibbs state), which is diagonal in the energy basis. Therefore, the entropy *S* of the system is simply $S=-\sum_{s}P_{s}\ln\left(P_{s}\right)$, which is reduced to the thermodynamic entropy when the probabilities are calculated in equilibrium, i.e., where the temperature is defined at each point of the cycle.

On the other hand, the classical case does not require a strong conservation restriction of the population in Equation (Equation 3), giving the possibility to have variations in thermal occupations along the adiabatic pathways. Furthermore, as the systems follow the classical thermodynamic formulation, they are kept in equilibrium in all positions of the S−B diagram, and the temperature can be defined at points *B* and *D*. A possible way to do this is to solve the total entropy differential equation dS(T,B)=0 to obtain information about the relationship between *T* and *B* along the isentropic processes. The first-order differential Equation is given by the following:(5)$$\begin{array}{c} \frac{dB}{dT}=-\frac{{\left(\frac{\partial S}{\partial T}\right)}_{B}}{{\left(\frac{\partial S}{\partial B}\right)}_{T}}. \end{array}$$

Another possibility is simply to impose the classical adiabatic condition between the points A–B and C–D in the form of
(6)$$\begin{array}{c} S\left(T_{l},B_{l}\right)=S\left(T_{B},B_{h}\right),\;S\left(T_{h},B_{h}\right)=S\left(B_{l},T_{D}\right). \end{array}$$

Because in the classical case the working substance is always in thermal equilibrium, the internal energy U(T,B) derived from the canonical partition function (Z(T,B)) is always well defined, i.e., $U=T^{2}\frac{\partial \ln(Z)}{\partial T}$. Accordingly, the incoming heat and released heat can be rewritten for the classical case as
(7)$$\begin{array}{c} Q_{in}^{cs}=U_{C}\left(T_{h},B_{h}\right)-U_{B}\left(T_{B},B_{h}\right), \end{array}$$
(8)$$\begin{array}{c} Q_{out}^{cs}=U_{A}\left(T_{l},B_{l}\right)-U_{D}\left(T_{D},B_{l}\right). \end{array}$$

According to these two expressions, the total work extracted in the classical formulation is given by
(9)$$\begin{array}{ccc} W^{cs} & = & \left[U_{C}\left(T_{h},B_{h}\right)+U_{A}\left(T_{l},B_{l}\right)\right]-\left[U_{B}\left(T_{B},B_{h}\right)+U_{D}\left(T_{D},B_{l}\right)\right], \end{array}$$
in which we have separated the points where the working substance comes into contact with thermal reservoirs (points A and C) and the other points assumed in thermal equilibrium for the classical formulation (points B and D).

The efficiency of classical and quantum cases is defined accordingly in the following way:(10)$$\begin{array}{c} \eta^{q}=\frac{W^{q}}{Q_{in}^{q}},\;\eta^{cs}=\frac{W^{cs}}{Q_{in}^{cs}}. \end{array}$$

At this point, we have obtained the quantum and classical expressions for the total work, as in Equations (Equation 4) and (Equation 9), respectively. To compare these two scenarios, we rewrite the quantum work in Equation (Equation 4) as follows:(11)$$\begin{array}{c} W^{q}=\left[U_{C}\left(T_{h},B_{h}\right)+U_{A}\left(T_{l},B_{l}\right)\right]-\left[U_{B}^{*}+U_{D}^{*}\right], \end{array}$$
where $U_{B}^{*}=\sum_{s}E_{s}\left(B_{h}\right)P_{s}^{B}$ and $U_{D}^{*}=\sum_{s}E_{s}\left(B_{l}\right)P_{s}^{D}$ are two expected values of energy in nonthermal equilibrium. This is where we can get a first relevant discussion to compare the classical and quantum approaches in Equations (Equation 9) and (Equation 11), respectively. According to thermodynamics, a system in equilibrium has the minimum value of energy for a given entropy. If this were not so, we can think that we could withdraw energy from the system (for example, in the form of work), keeping the value of the entropy constant, and then, we could return this energy to the system in the form of heat [24]. This would leave the system with its initial state of energy but would cause the entropy to increase, violating the condition that, for a state in equilibrium, the value of the entropy corresponds to a maximum [24]. Therefore, inspecting Equations (Equation 9) and (Equation 11), we can argue that the quantity $U_{B}^{*}+U_{D}^{*}$ is always greater than $U_{B}\left(T_{B},B_{h}\right)+U_{D}\left(T_{D},B_{l}\right)$ because B and D points in the quantum Otto cycle are in nonthermal equilibrium. As such, we can conclude that the classical work will always by greater than or equal to the quantum work because the first two terms in Equations (Equation 9) and (Equation 11) are the same. Consequently, we can write the condition for total work extraction as
(12)$$\begin{array}{c} W^{cs}\geq W^{q}. \end{array}$$

This is a clear example of the robustness of thermodynamics. The result presented in Equation (Equation 12) is nothing more than the condition that a reversible total work is always greater than or equal to the irreversible one:(13)$$\begin{array}{c} W_{reversible}\geq W_{irreversible}. \end{array}$$

In the next subsections, we show the analysis for two cases, a two-level and a three-level system. We begin with a toy two-level model in order to present a simple situation where classical and quantum approaches give analytical expressions for the thermodynamic quantities previously described. Even though this scenario its quite simple, an important conclusion is drawn regarding the condition of work in Equation (Equation 12). By using this result, we study a three-level quantum dot-like system and explore first what parameter conditions must be considered to achieve equal works in Equation (Equation 12) and, second, how to maximize the work difference between the classical and quantum cases.

## 3. The Case of a Two-Level System

In this section, we address the case of a two-level system to guide the discussion towards a classical and quantum comparison of work and efficiency in the Otto magnetic cycle. For this purpose, let us consider a working substance described by two energy levels, $E_{1}\left(B\right)$ and $E_{2}\left(B\right)$, which are continuous and differentiable as a function of magnetic field. As usual, the thermal populations are given by the following:(14)$$\begin{array}{c} P_{1}\left(T,B\right)=\frac{e^{-\beta E_{1}\left(B\right)}}{Z(T,B)},\;P_{2}\left(T,B\right)=\frac{e^{-\beta E_{2}\left(B\right)}}{Z(T,B)}, \end{array}$$
where $Z\left(T,B\right)=e^{-\beta E_{1}}+e^{-\beta E_{2}}$ and $\beta =\frac{1}{T}$ (with $k_{B}=1$). The thermal populations satisfies the normalization condition $P_{1}\left(T,B\right)+P_{2}\left(T,B\right)=1$. In this case of just two energy levels, we can write the von Neumann’s entropy as
(15)$$\begin{array}{c} S=-P_{1}\ln\left(P_{1}\right)-P_{2}\ln\left(P_{2}\right). \end{array}$$

If we associate this entropy with the one derived from thermal equilibrium, it is possible to obtain the differential equation that relates changes between temperatures and magnetic field along an isentropic path. Taking the total difference of Equation (Equation 15), we get
(16)$$\begin{array}{c} \frac{dT}{dB}=\frac{T\left(\frac{dE_{2}\left(B\right)}{dB}-\frac{dE_{1}\left(B\right)}{dB}\right)}{E_{2}\left(B\right)-E_{1}\left(B\right)}, \end{array}$$
for which the solution is given by
(17)$$\begin{array}{c} CT\left(B\right)=\left(E_{2}\left(B\right)-E_{1}\left(B\right)\right), \end{array}$$
where C it is an integration constant. Equation (Equation 17) is a general solution for the temperature regardless of the behavior of the energy levels upon the external field *B*. Replacing this solution as the thermal populations in Equation (Equation 14), we found $P_{1}$ and $P_{2}$ to be only C dependent rather than *B* or *T* in the following form:(18)$$\begin{array}{c} P_{1}=\frac{e^{C}}{1+e^{C}},\;P_{2}=\frac{1}{1+e^{C}}, \end{array}$$

Figure 2 shows the probabilities of Equation (Equation 18) and the entropy of Equation (Equation 15), where for C→0 (high-temperature limit), the entropy (blue line) reaches their maximum value of ln(2) and, for the case of C→∞ (low-temperature limit), $P_{1}$ goes to one and $P_{2}$ goes to zero as expected. An important consequent of this C dependence can be see analyzing the adiabatic paths in the Otto cycle of Figure 1. Due to the fact that $P_{1}$ and $P_{2}$ do not vary with *T* and *B*, their values are constant along the adiabatic processes; therefore, both classical and quantum works become the same in Equation (Equation 12) because the quantum restriction on the population its automatically fulfilled. This result does not depend on how $E_{1}$ and $E_{2}$ behave upon *B*. With this, we can conclude that, for every two-level system in which the energies are continuous with *B*, there will be no difference on the mathematical analysis between the classical and quantum approaches except for conceptual interpretation about the physical processes.

In this same line of discussion, we found another case for W${}^{cs}$=W${}^{q}$ (Equation (Equation 12)) corresponding to the instance when the working substance in the classical adiabatic strokes satisfies the differential equation in the following form:(19)$$\begin{array}{c} \frac{dT}{dB}\propto \frac{T}{B}, \end{array}$$
for which the trivial solution corresponds to a linear relation between the magnetic field and temperature, that is $T\left(B\right)=C_{1}B$, where $C_{1}$ is an integration constant. Take as example a system for which the energy levels are mathematically described by $E\left(B\right)={\left(-1\right)}^{j}jB$, where *j* can take integer values from zero onwards; the thermal populations and the partition function for this system can be defined as follows:(20)$$\begin{array}{c} P_{j}=\frac{e^{\frac{{\left(-1\right)}^{j+1}jB}{T}}}{Z(T,B)},\;Z\left(T,B\right)=\sum_{j}e^{\frac{{\left(-1\right)}^{j+1}jB}{T}}. \end{array}$$

Using the same reasoning as before, it can be deduced that this kind of system will have a linear dependence as in Equation (Equation 19) between *T* and *B* during the adiabatic phase (i.e., T∝B). Replacing this solution in Equation (Equation 20), we recover the same constant population behavior as before, giving as a consequence of the equality between the quantum and classical work Equation (Equation 12).

## 4. The Case of a Three-Level System

In order to show a case in which the quantum and classical work can coincide, we will take as an example a three-level system for which the spectrum in Figure 3a simulates (roughly) a graphene quantum dot under an external magnetic field [23]. The chosen levels, all in arbitrary units, are given by $E_{1}=0$, $E_{2}=B$ and $E_{3}=e^{-B}$. This spectrum of energy exhibits a crossing between the levels $E_{2}$ and $E_{3}$ for B>0 and is doubly degenerated for zero external magnetic field.

The thermodynamics quantities are calculated from the partition function $Z=1+e^{-\frac{B}{T}}+e^{-\frac{e^{-B}}{T}}$, and the entropy *S* as a function of external magnetic field is presented in Figure 3b. At high temperature and low-external magnetic field region, the entropy tends to ln(3) because, at high temperature, all three energy levels have the same occupation and therefore the same probability. From the same figure, we note that some combinations of parameters at low temperature produce first a decrease of entropy for external magnetic field B<2 and then an increase converging to ln(2) for high magnetic fields. This is due to the form of the $E_{3}$ level in Figure 3a, since for high magnetic fields, it begins to be easier to populate it than the $E_{2}$ level, which corresponds to a state that grows linear with the value of the external field. Consequently, in a region of high magnetic field and low temperature, we have only two populated states, obtaining the value of ln(2) for the entropy again.

To discuss the performance of the classic and quantum Otto cycles for the three-level system, we have selected two zones where we will apply the cycles displayed in the Figure 3b and in the diagram of temperature versus external field in Figure 4, obtained by using the classical conditions of an isentropic process described by the Equation (Equation 6). The green geometric shape (zone 1) corresponds to a zone where the magnetic field and temperature are related to each other in a linear way, and the purple shape (zone 2) is a region where the temperature and the external field have a nonlinear relation between them. The work and efficiency for zone 1 is presented in Figure 5 for $T_{h}=2$, $T_{l}=0.75$ with an starting value of the external field given by $B_{l}=3.00$ and the high magnetic field ($B_{h}$) is moving up to the value of 4.5. As we can see in the figure, both work and efficiency are the same regardless of the classical or quantum natures of the cycle operation. How can this be explained? A possible answer to this question is found in the *T*-*B* diagram in Figure 4. If we recall the discussion from the previous two-level system section, it was found that there are two possible situations where classical and quantum works are the same. The first one occurs when the system has only two energy levels with any magnetic field dependence. Here, a constant population behavior appears that implies a fulfillment of the quantum’s restrictions, giving as a consequence the equality of work and efficiency with their classical counterpart. The second case was a particular situation in which all energy levels have a linear dependence with the external field. The result of this is a linear relationship between *T* and *B* appearing for the adiabatic strokes, and therefore, we recover the same population behavior of a two-level system. Therefore, both classical and quantum work are the same even if the system has a multilevel spectrum (where the B−T relation is linear).

In the present case, due to the set of chosen parameters, a linear dependence T−B is graphically found in Figure 4. What this tell us is that, even if the system does not have two levels or a spectrum with a linear magnetic field dependence, if there exists a zone in which a linear-like relationship can be found between *T* and *B*, the system will present a maximum quantum work. In other words, the quantum work will be equal to the classical one while the thermal machine operates inside that parameter set.

If we analyse the work and efficiency for zone 2 in Figure 6, where the parameters are given by $T_{l}=0.30$ and $T_{h}=1.40$ and the external field is in the range of 0.75 up to 1.50, we observe a drastic decrease in the quantum work and efficiency for points near the initial field value $B_{l}$. Two things happen in that area which are fundamental to obtain these results. The first one is the fact that the function connecting the A–B and C–D paths in the *T*–*B* diagram (adiabatic) in Figure 4 clearly does not have a linear relation T−−B. This implies that at least one of the two situations where the classical and quantum works are the same is not fulfilled, as described for the two-level system. A second observation can be made if we look at Figure 3a for the range of magnetic field involved in the process (zone 2). In this region, we see that the three levels play an important role in determining the thermal populations, and therefore, it cannot behave as a two-level system as the cycle operating the green zone 1 of the *T*–*B* diagram in Figure 4. These results confirm that the inclusion of more energy levels causes a lower work and efficiency in the quantum Otto Cycle when the working zone on the *T*–*B* diagram localizes in regions where no linear relationship between *T* and *B* can be found as well as when more than two levels play a significant role in the thermal populations.

We want to emphasize that the discussion of the two cycles presented in this work correspond to idealized cycles in which the output power of the machine is zero. In order to realize a more feasible Otto cycle for practical implementation protocol, this work can be extended in the shortcuts to the adiabaticity technique [25,26,27]. Additionally, the use of quantum resources (such as quantum coherence, for example) can lead to an enhancement in the performance of the quantum version of the proposed cycle.

## 5. Conclusions

In this work we have studied and compared Otto’s classical and quantum cycles, showing that the amount of extracted work under classical formulation is equal to or higher than its quantum counterpart. An explanation for this lies in all-timing equilibrium configurations along the cycle operation in the classical one, while in the quantum case, the only equilibrium states belong to the thermal reservoirs. This is a general result valid for any working substance that operates under the restriction mentioned in the first section of this work due to the fact that, in a constant entropy process, the lowest energy state is achieved at thermal equilibrium. Furthermore, this result is consistent with the thermodynamic postulate of maximum work since the Otto quantum cycle, having just two of the four main states at thermal equilibrium, represents an irreversible cycle in comparison to the classical one in which all possible states belong to equilibrium configurations. We also studied two particular cases where the classical and quantum efficiency/work are the same. This corresponds to any two-level system and to the case where a multilevel system is linearly dependent on the magnetic field and the energy spectrum and where a linear relationship between temperature and magnetic field can be found during adiabatic stages. Finally, we corroborate these results by extending the analysis to a three-level system. When just two of the three-levels become relevant during the population thermal evolution, a linear relationship *T*–*B* is recovered (adiabatic processes) and, therefore, an equality between classical and quantum work is achieved. If the external parameters (reservoir’s temperatures, external magnetic fields, etc.) are such that all three levels becomes relevant, we lose the equality and a nonlinear relation arises between temperature and magnetic field during the adiabatic stages, giving a lower quantum work extraction in relation to the classical one. 

## Figures and Tables

**Figure 1 entropy-22-00755-f001:**
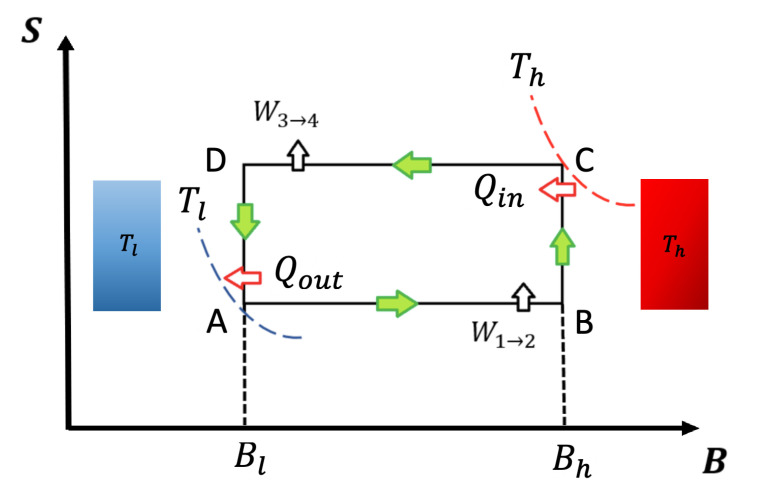
Entropy versus external field diagram for classical and quantum Otto cycle: The system contacts the thermal reservoirs only in the isomagnetic strokes. At the points C and A, the working substance reaches the temperatures $T_{h}$ and $T_{l}$, respectively, indicated by the isotherms touching the points. For the quantum version, the entropy values $S_{B}$ and $S_{D}$ are calculated using the same thermal probabilities as in points A and C to ensure the quantum adiabatic strokes A→B and C→D.

**Figure 2 entropy-22-00755-f002:**
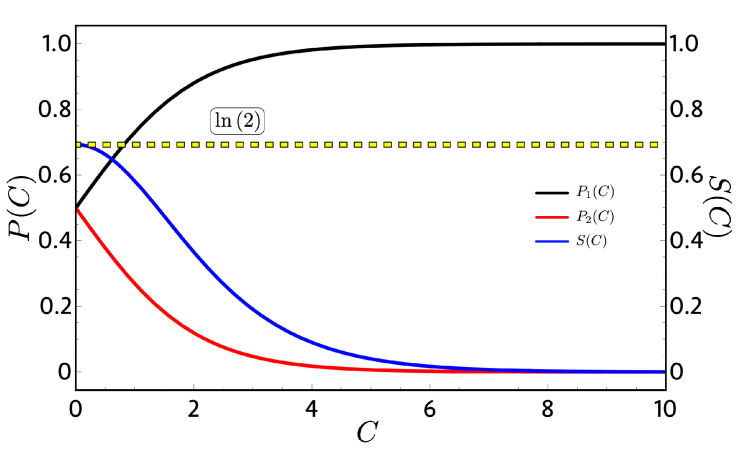
Probabilities (P) (left vertical axis) and entropy (S) (right vertical axis) as a function of the integration constant C: We observe that, when the integration constant is zero (high-temperature limit), we obtain the maximum value of the entropy given by ln(2) because the probabilities take the value of 1/2 at each level.

**Figure 3 entropy-22-00755-f003:**
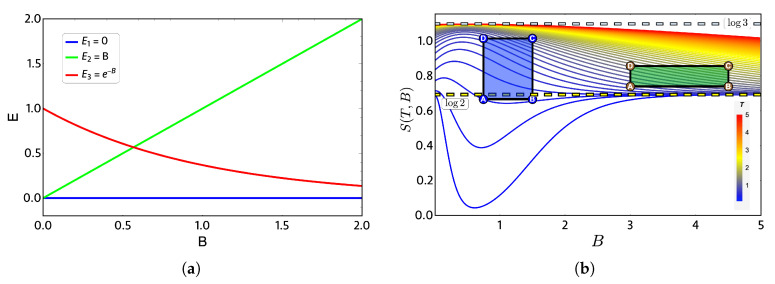
(**a**) Proposed three-level energy spectrum as a function of external dimensionless magnetic field parameter *B*. (**b**) Entropy diagram (*S*) versus external field (*B*) in arbitrary units: the color bar indicates high (low) temperature as a red (blue) gradient. Green and purple geometric shapes represent two zones: 1 and 2, respectively, where the Otto cycle of Figure 1 is applied in its classical and quantum versions. Notice that the yellow (blue light) dashed horizontal lines indicate entropy constant values of ln(2) (ln(3)).

**Figure 4 entropy-22-00755-f004:**
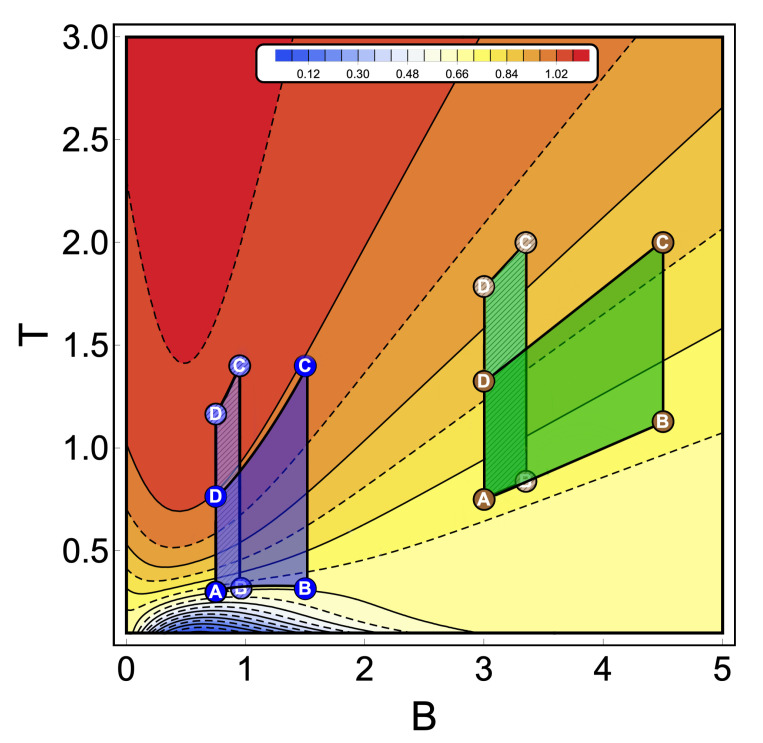
Temperature versus external field diagram obtained when the classical conditions over the isentropic trajectories are applied to the three-level system. The color scheme represents different constant values of entropy (in arbitrary units) from 0.04 (blue) up to 1.09 (red). From this diagram, the green geometric shapes represent a zone where the temperature is linear with the magnetic field. The purple one is a region where the relation between temperature and the magnetic field has a nonlinear behavior. A dashed pattern is used to represent two different thermal cycles operating inside the same zone keeping in both cases the same reservoir’s temperatures (A and C between dashed and non-dashed per area). These particular cycles are indicated in Figure 5 and Figure 6, respectively.

**Figure 5 entropy-22-00755-f005:**
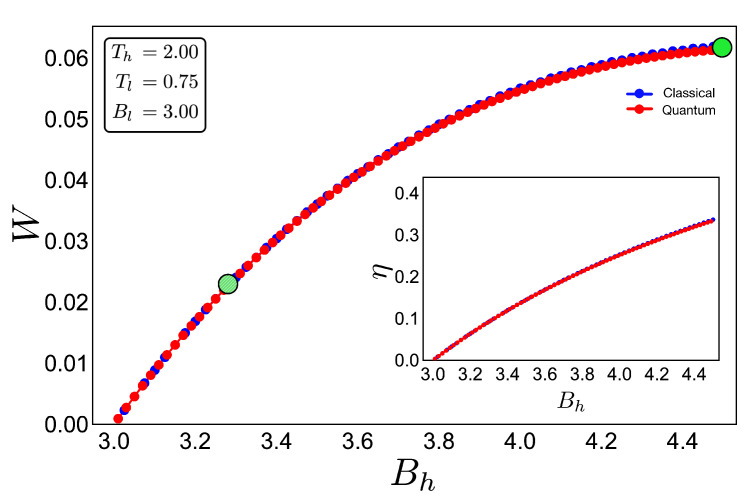
Work (in arbitrary units) and efficiency (inset) for the case of zone 1 (green shape in Figure 4) for an initial value of external magnetic field $B_{l}$= 3.00 and up to $B_{h}$ = 4.50: The hot and cold reservoirs are $T_{h}$ = 2.00 and $T_{l}$ = 0.75, respectively. The blue dotted line represents the classical performance, and the red dotted line represents the quantum one. Two bigger points, green and dashed-green, represent the two geometric shapes (cycles) showed in Figure 4.

**Figure 6 entropy-22-00755-f006:**
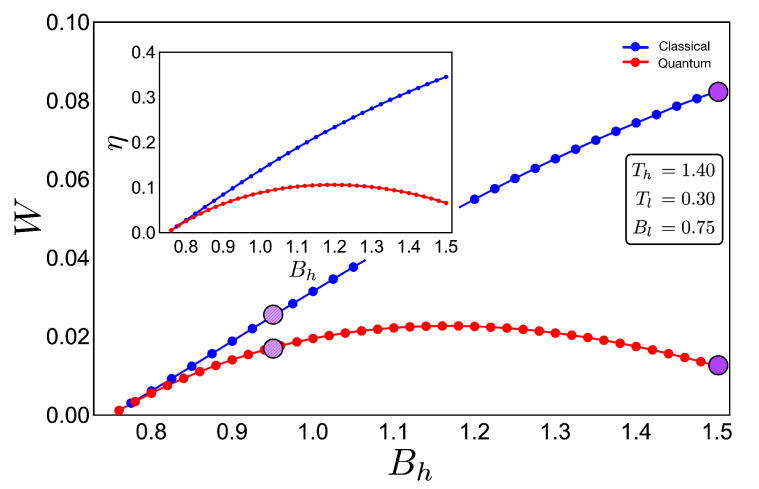
Work (in arbitrary units) and efficiency (inset) for the case of zone 2 (purple shape in Figure 4) for an initial value of external magnetic field $B_{l}$ = 0.75 and up to $B_{h}$ = 1.50:The hot and cold reservoirs are $T_{h}$ = 1.40 and $T_{l}$ = 0.30, respectively. The blue dotted line represents the classical performance, and the red dotted line represents the quantum one. Four purple circles (two per cycle) are drawn to show the difference between cycles (see Figure 4) inside the same region. It can be seen that the purple line with marked circles that are close together indicate a similar performance for the classical and quantum cases due to the increasingly linear function in the adiabatic path for the purple line marked cycle shown in Figure 4.

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
