# Peer review of "Otto Engine: Classical and Quantum Approach"

_entropy, 2020, doi:10.3390/e22070755_

Round 1

Reviewer 1 Report

This manuscript examines and compares the total work extracted from the quantum and classical versions of the magnetic Otto cycle. The authors argue that the work obtained from the classical engine is always greater than or equal to that obtained from the quantum version.

They illustrate in the case of a two-level system that the work obtained from the classical and quantum versions is the same when there exists a linear relationship between the temperature and magnetic field. This also holds in the analysis of a three-level system, whereas the work extracted from the quantum case is lesser than that of the classical one when a non-linear relation exists between temperature and magnetic field.

The manuscript is well-written and well-presented. It is also clear in its goals and results. I have some minor comments for the authors' consideration.

1. p. 3. What are the implications of the use of a diagonal density matrix in their approach? Is is just a convenient selection? To connect to the classical case?

2. p. 4. I could not follow the reasoning of how Eq. (10) is obtained from Eq. (4). Is it an attempt to formulate the quantum case in classical terms for interpretational purposes? The authors could be more explicit on this point. The relation for the last term of Eq. (10) could be provided.

3. Figs. 5 and 6 provide plots of the efficiency and an accompanying discussion, however the efficiency has not been defined in the manuscript.

4. Is there a physical interpretation for the presence of the maxima (work and efficiency) in Fig. 6 for the quantum case? Should this be discussed?

Author Response

Dear Referee, please see the attached file.

Very thankful, The authors.

Reviewer 2 Report

The authors analyze the total work extracted and the efficiency of the magnetic thermodynamic cycle in its classic and quantum versions. The general result of this analyze is the conclusion that the work and efficiency of the classical engine is always greater than or equal to its quantum counterpart, independent of the working substance. I think that this conclusion is wrong since some cases of the magnetic thermodynamic cycle in its quantum versions are not conidered in the manuscript. The authors have not considered the magnetic thermodynamic cycle in which the persistent current appears because of the quantuzation and disappears because of energy dissipation in superconducting rings plased in magnetic field. It was demonstrated experimentally in the publication [1] that that useful work can be obtained from heat with the help of this thermodynamic cycle. The manuscript should not be published until the authors consirde the efficiency of the quantum heat engine demonstrated in [1] and prove that its efficiency is less than that of the classical engine.

[1] V.L. Gurtovoia, V.N. Antonov M. Exarchos, A.I. Il’in, A.V. Nikulov, The dc power observed on the half of asymmetric superconducting ring in which current flows against electric field. Physica C: Superconductivity and its applications. 559, 14-20 (2019).

Author Response

(The authors gave the same response as above.)

Reviewer 3 Report

The authors develop some details about the Otto cycle engine using magnetic models for the quantum and classical cases. They provide guidance on how to optimize the efficiency and power of these Otto engines. The manuscript seems interesting, and provides specific results of relevance. I recommend publication after the authors have addressed the following concerns.

In addition to several grammatical corrections (some of them listed below), the main technical issue I have is with the classification of the cycles shown in Fig. 4. The two boxes shown are not rectangles. The green box could be a trapezoid, but I suspect the adiabatic steps are not precisely linear. The authors should take care to correctly classify the shapes of the boxes and the field dependence of the temperature.

Some grammatical corrections. Please carefully proofread the manuscript for others.

Line 28, both admit changes…system through heat exchange

Line 29, energy-level populations…

Line 148, move k_B up one sentence. (k_B=1 is also used in the previous sentence).

Figure 4 caption: green rectangle (maybe a trapezoid?), temperature is nearly linear with field

Line 213, indent.

Line 223, clearly does not have…

Line 227, cannot (best not to use contractions in scientific writing)

Line 235, only equilibrium states belong to

Line 240, since the Otto quantity

Line 254, relevant to developing a quantum engine; (The final paragraph could use additional clarification, its meaning is not clear.)

Author Response

(The authors gave the same response as above.)

Reviewer 4 Report

The presented manuscript is devoted to an interesting field, i.e. classical and quantum Otto engines. By making use of the quantum systems with a few (two and three) energy levels the authors show that the total work of the "classical" engine is larger or equal than the "quantum" engine. Moreover, for the systems with three energy levels the two processes were considered: the temperature varies linearly (nonlinearly) with magnetic field.  In the case of the nonlinear dependence the "classical" Otto engine outperforms the “quantum” one.  The authors explained this effect by the presence of non-equilibrium  vs. equilibrium states for the quantum (classical) engine. 

I  find that the results are valid and  the paper can be interesting for scientists working on the thermodynamics and transport of various artificially prepared quantum systems. I think that paper can be improved if the authors implement in the manuscript the discussion how to experimentally realize the classical and , especially, quantum engines, and how to measure the total work of the device. There are also a few typos, e.g. "especific" (page 3, line 64), Fig. 2 -log 2 instead of $ln 2$ etc.

I recommend the manuscript to the publication in the Entropy journal with minor (optional) changes. 

Author Response

(The authors gave the same response as above.)

Reviewer 5 Report

Before taking the decision about the publication and further revision, I would like to clarify one point in the paper.

Authors provide the Eq.(16) for temperature vs magnetic field. Suppose, I set the constant C=0, then T goes to 0, or \beta=1/T goes to infinity. Substituting now \beta into the Eq.(13) gives 1) P1 goes to 0, if E1>E2; 2)P1 goes to 1, if E1<E2. However, from the Fig.2 at C=0, one has P1=1/2. What does that mean? To continue, if one substitutes the Eq.(16) into Eq.(13), then in the exponents of Eq.(17) it should be 1/C, not C. Is it C or 1/C? 

Author Response

(The authors gave the same response as above.)

Round 2

Reviewer 2 Report

The main conclusion of the manuscript is that the work and efficiency of the classical engine is always greater than or equal to its quantum counterpart, independent of the working substance. This conclusion is wrong since the authors do not consider all cases of the magnetic thermodynamic cycle in its quantum versions. The manuscript is misleading. Therefore I recommend to reject this manuscript.

Reviewer 5 Report

 Accept in present form